# 🛠️ CRAFT: Customizing LLMs by Creating and Retrieving from Specialized Toolsets

**Lifan Yuan**,[*] **Yangyi Chen**[*], **Xingyao Wang, Yi R. Fung, Hao Peng, Heng Ji**
University of Illinois Urbana-Champaign
{lievanyuan173}@gmail.com
{yangyic3,xingyao6,yifung2,haopeng,hengji}@illinois.edu

## Abstract

Large language models (LLMs) are often augmented with tools to solve complex tasks. By generating code snippets and executing them through task-specific Application Programming Interfaces (APIs), they can offload certain functions to dedicated external modules, such as image encoding and performing calculations. However, most existing approaches to augment LLMs with tools are constrained by general-purpose APIs and lack the flexibility for tailoring them to specific tasks. In this work, we present **CRAFT**, a general tool creation and retrieval framework for LLMs. It creates toolsets specifically curated for the tasks and equips LLMs with a component that retrieves tools from these sets to enhance their capability to solve complex tasks. For each task, we collect specific code solutions by prompting GPT-4 to solve the training examples. Following a validation step ensuring the correctness, these solutions are abstracted into code snippets to enhance reusability, and deduplicated for higher quality. At inference time, the language model retrieves snippets from the toolsets and then executes them or generates the output conditioning on the retrieved snippets. Our method is designed to be flexible and offers a plug-and-play approach to adapt off-the-shelf LLMs to unseen domains and modalities, without any finetuning. Experiments on vision-language, tabular processing, and mathematical reasoning tasks show that our approach achieves substantial improvements compared to strong baselines. In addition, our in-depth analysis reveals that: (1) consistent performance improvement can be achieved by scaling up the number of tools and the capability of the backbone models; (2) each component of our approach contributes to the performance gains; (3) the created tools are well-structured and reliable with low complexity and atomicity. [1]

## 1 Introduction

Large language models (LLMs) have emerged as transformative tools in AI, exhibiting capabilities in complex problem-solving, including reasoning, planning, and producing creative outputs (Brown et al., 2020; Touvron et al., 2023b;a; Yuan et al., 2023). Recent evidence has shown that LLMs can dynamically interact with the environment through external tools, which grants them access to information beyond their pretrained parameters (Qin et al., 2023a; Mialon et al., 2023; Schick et al., 2023). For example, these models can generate code snippets and call APIs provided by visual tools like image encoding models, to solve problems that involve images or videos (Wu et al., 2023; Shen et al., 2023; Yang et al., 2024).

Success has been achieved by integrating LLMs with large-scale, general-purpose tool collections (Qin et al., 2023b; Tang et al., 2023; Surís et al., 2023; Gao et al., 2023a; Chen et al., 2022a; Gao et al., 2023b; Patil et al., 2023). However, adapting LLMs to many domains and evolving applications involves working with more specialized APIs tailored to address specific challenges, which are often inadequately represented in general-purpose toolsets. In response, this work proposes to integrate LLMs with highly customizable toolsets that are curated for specific problems of interest.

---

[*]Equal contribution. The first author conducts this research during an internship at UIUC.
[1]The code is available at `https://github.com/lifan-yuan/CRAFT`.

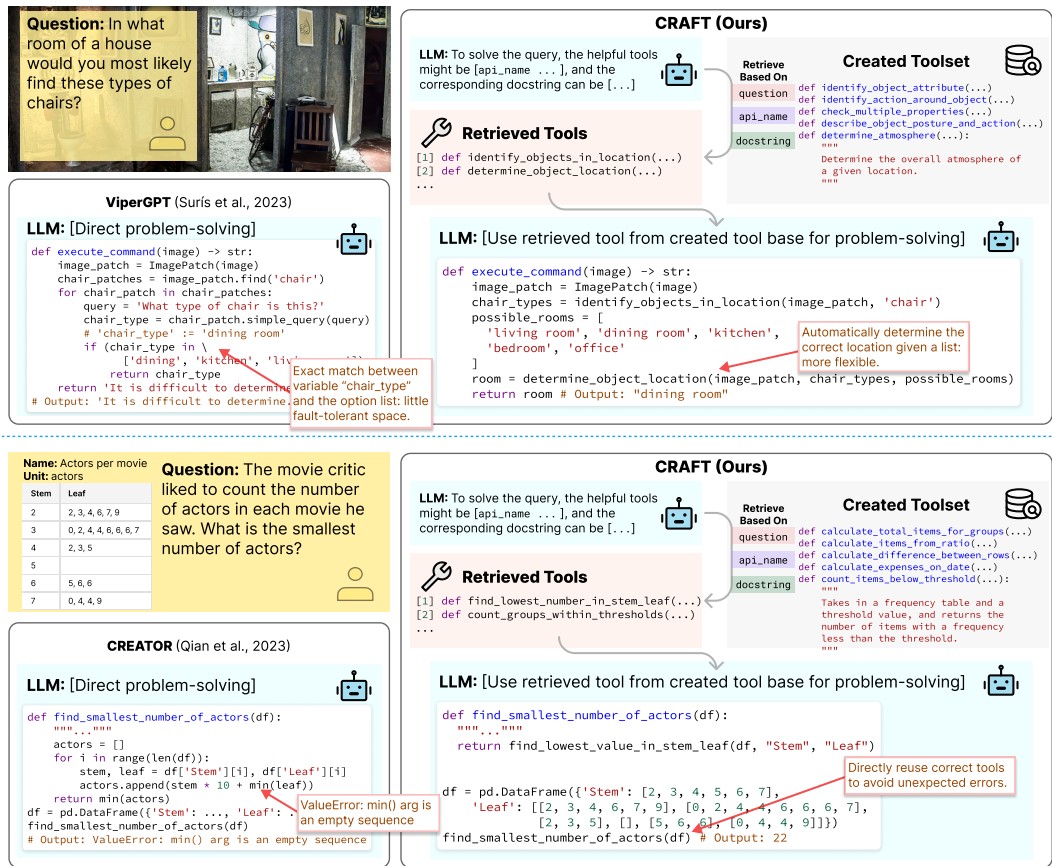

Figure 1: Previous approaches directly solve the given problem by generating code solutions, which may contain errors. CRAFT first creates a toolset that contains diverse, reusable, and correct tools that are executable code snippets. During inference, CRAFT employs a multi-view matching approach, incorporating information about the target problem, API names, and docstrings, to identify and utilize relevant tools, enhancing its problem-solving capabilities.

Our approach, dubbed CRAFT, constructs a toolset customized for a given task (see Figure 1). In contrast to previous approaches that only incorporate one single type of tool (Cai et al., 2023) or create unverified and non-reusable tools (Qian et al., 2023), our toolset contains diverse, reusable, and correct APIs that can tackle various problems. This is achieved through an automated process, by instructing LLMs to generate specific code solutions to solve training problems of the task or related ones. The specific solutions are then abstracted into code snippets, which can later be instantiated to solve similar problems. Dedicated validation and deduplication steps ensure the correctness of the tools and reduce redundancy, thereby enhancing the quality of the toolset.

At inference time, precisely identifying and retrieving relevant tools for the given problems is challenging, especially given the constructed large toolset. Existing solutions typically rely on pre-selected tools (Parisi et al., 2022), heuristic-based tool selection strategy (Shen et al., 2023), and simple similarity measure (Qin et al., 2023b), which may be unsuitable or insufficient to pinpoint the related tools from a large toolset given the problems. CRAFT implements a retrieval component that takes into account the target problem, the names of the tools (a.k.a, APIs), and their docstrings through a multi-view matching function. The retrieved snippets are then added to the prompt of LLMs so that the retrieved tools can be invoked in the generated code solutions.

The empirical effectiveness of CRAFT is validated through experiments on visual question answering, tabular processing, and mathematical reasoning tasks. Compared to strong baselines, CRAFT achieves an average of 43.16% relative improvement in F1 score compared to the best baselines in vision-language tasks, where the LLMs are required to interact with various visual tools to encode the images. Through our carefully designed analysis, we find (1) the performance continually increases as the number of tools and the capability of the backbone models increase; (2) Each component

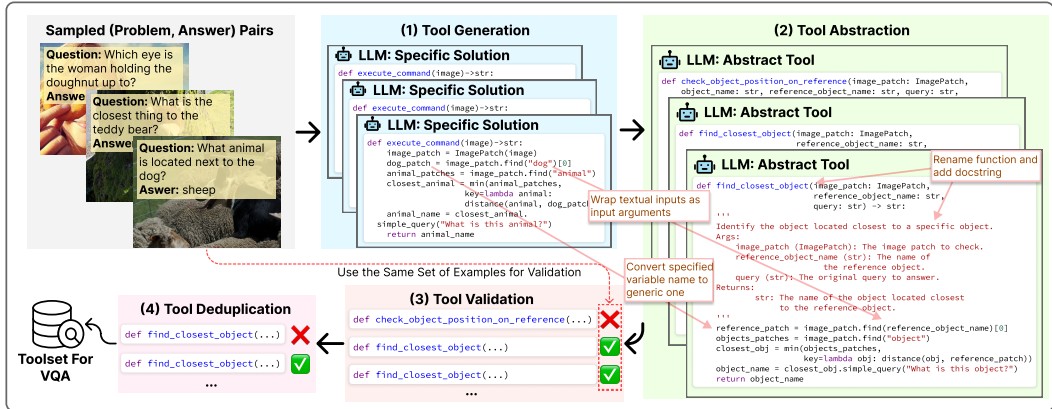

Figure 2: The toolset construction pipeline creates diverse, reusable, and correct tools that are executable code snippets, which can generalize LLMs to specialized domains and tasks.

design incorporated in CRAFT contributes to the performance gains; (3) the created tools exhibit atomicity and possess low complexity, underscoring their robust structures and reliability.

The contribution of this work is two-fold. First, we introduce CRAFT, a broadly applicable framework to customize LLMs to various tasks and domains via tool creation and retrieval. Second, we release the created toolsets that include diverse, reusable, and correct tools, which are useful for various downstream tasks. Estimatedly, it costs around 2,500$ in total for the toolsets construction.

# 2 🧑‍🍳 CRAFT

We introduce CRAFT to address the challenges faced by prior research in the following two aspects: (1) **Tool Creation:** The establishment of an extensive toolset of diverse, reusable, and correct tools, in contrast to the reliance on limited examples (Cai et al., 2023; Qian et al., 2023); (2) **Tool Retrieval:** The effective retrieval of relevant tools from a large toolset, tailored to the specific question, thereby departing from the conventional approach of simplistic similarity matching (Qin et al., 2023b; Patil et al., 2023). By instantiating the retrieved code and adding it to the prompt, LLMs can then use the tools by calling the function to perform complex operations rather than implement every detail from scratch.

## 2.1 TOOL CREATION

Based on a source dataset, namely a general instruction dataset or a training dataset that contains problem-answer pairs, CRAFT constructs the toolset through four steps: **Generation**, **Abstraction**, **Verification**, and **Deduplication**, which are illustratied in Figure 2 and will be described as follows.

**Generation.** To create a toolset containing diverse tools that can be adopted to address various problems, we apply an iterative approach to sample problem-answer pairs from the source dataset. At a high level, the generation step involves iteratively sampling problems from the source dataset, generating code solutions, and filtering out incorrect ones. We use $Q$ to denote the set of sampled problems and $R_i$ to denote the set of remaining problems after the $i$-th iteration. $Q$ is initialized with $n$ random samples from the entire source dataset and $R_i$ is initialized as the rest. At each iteration, we use the highest similarity between each $q_r \in R_i$ and any $q_s \in Q$ as the similarity between each $q_r$ and set $Q$. To enhance the diversity of the toolset, $Q$ is updated by adding $k$ problems that are least similar to $Q$, where $k$ represents the desired number of samples for each iteration. This min-max sampling strategy is: $Q \leftarrow Q \cup \mathrm{argTopK_{min}} \left( \max_{q_s \in Q} \mathrm{sim}(q_r, q_s) \mid q_r \in R_i \right)$. Function $\mathrm{argTopK_{min}}$ returns the indices of the top $k$ elements with the smallest values from a set, which is set to 100 in our implementation, and $\mathrm{sim}(\cdot)$ denotes the cosine similarity of the representation vectors computed by SimCSE, a state-of-the-art sentence representation learning method based on contrastive learning (Gao et al., 2021).

For each problem $q_r \in Q$, we instruct GPT-4 (OpenAI, 2023) to generate a specific solution in Python that can be executed by an interpreter to get the answer. The prompts are shown in Appendix C. We keep those code solutions that are bug-free and can produce correct outputs, and discard everything else to ensure the correctness of the created tools.

**Abstraction.** The generated code solutions are tailored for the given problems, keeping them from being useful for others. The abstraction step aims to promote the reusability of the toolset, ensuring that each tool can be adopted to tackle a broader range of similar problems. This abstraction step is achieved by instructing GPT-4 to replace all specific variable names with general ones (e.g., `cat`→`animal`, `desk`→`object`) and wrap textual inputs of internal function calls as arguments of the tool (e.g., `date = df["date"]`→`date = df[column_name]`, where the value of `column_name` is passed in by tool users) within the code piece, substituting them with more generic counterparts to adapt to similar problems (see Figure 2). In addition, we instruct GPT-4 to assign a suitable and general function name and compose a corresponding docstring to elucidate the functionality of created tools. The prompt is described in Appendix C.

**Validation.** The validation step ensures the correctness of the created tools. This is achieved by examining whether the abstract tool functions can solve the original problems. Specifically, we offer GPT-4 access to the abstract tool function, with the expectation that it will address the original problems by supplying appropriate arguments to the tool function. The tools that fail to derive the correct answers given the original problems are discarded.

**Deduplication.** To reduce the redundancy in the toolset and improve its diversity, we perform a deduplication step to streamline the toolset and mitigate potential confusion stemming from redundant tools (e.g., same function names). We organize created tools into groups based on function names and the corresponding number of input arguments. Each group contains tools that have the same function names and the number of input arguments. For groups that contain more than one tool, we prompt GPT-4 to decide on the most comprehensive tool with extensive applications within the groups, using the prompt shown in Appendix C.

## 2.2 Tool Retrieval

Retrieving relevant tools from the large constructed toolset is challenging. For better retrieval outcomes, we prompt the LLM to "describe what it needs". During inference, the evaluated LLM is asked to generate the function names $f_t$ and the docstrings $d_t$ based on the target problem $q_t$. Then CRAFT adopts a similarity measuring strategy that takes into account three key aspects of the created tool $t_i$: (1) The original problem used for creating the tool $q_i$; (2) The tool's function name $f_i$; (3) The docstring of the function $d_i$. For each tool $t_i$, this results in a tuple $(q_i, f_i, d_i)$. We conduct multi-view matching, searching tools via $q_t$, $f_t$, and $d_t$ respectively in the toolset $T$. Specifically, we have:

$$T_{q_t} = \mathrm{argTopK}_{\max} \left( \mathrm{sim}(q_i, q_t) \mid t_i \in T \right) \tag{1}$$

where $\mathrm{argTopK}_{\max}$ is a function that returns the indices of the top $k$ elements with the maximum values from a set, $\mathrm{sim}(\cdot)$ measures the similarity between two sentences using SimCSE embeddings, and $T_{q_t}$ is a list of $k$ tools retrieved by matching problems. We then perform the similar retrieval by matching function names and docstring, obtaining $T_{f_t}$ and $T_{d_t}$ respectively. Next, the three lists of tools are aggregated and ranked by their frequency of occurrences. We then retrieve the three most frequent tools by majority vote. Finally, we filter out those that occur only once, if any. In extreme cases, it is also possible that all tools appear only once, i.e. the retrieved tool set is empty, then LLMs would directly perform code generation to solve question without invoking task-specific tools.

After retrieval, the code snippets of tools are added to the prompt of LLMs for code generation to solve a given question. LLMs can invoke the tools (a.k.a, APIs) embedded in the code. Subsequently, the retrieved tool functions and LLM-generated code solutions are instantiated into executable code, and then they are executed to obtain the final predictions.

**Summary and Discussion.** CRAFT creates a specialized toolset offline, and retrieves useful tools from the toolset in inference time. In toolset creation, we apply an iterative problem-sampling strategy based on similarity for diversity, followed by generating code solutions using GPT-4. To ensure the reusability of the created tools, we abstract the specific solutions into high-level tools that can tackle various kinds of problems by instructing GPT-4. To ensure the tools' correctness, we evaluate the tools on the original problems and discard those outputting incorrect answers. Finally, we deduplicate

the tools to reduce redundancy, and finally obtain a toolset. In inference, we apply a multi-view matching algorithm regarding the target problem, function name, and docstring between those in the toolset to retrieve related tools.

We highlight several advantages of CRAFT. At a high level, by leveraging the tool creation paradigm, we can effectively utilize the domain-specific data to customize the LLMs without extensive fine-tuning, rendering CRAFT a **training-free** and **plug-and-play** approach. Due to CRAFT's flexibility in accommodating various domains and tasks, it is **broadly applicable** across a spectrum of problem categories. In the concrete implementation, each tool is instantiated as an executable code snippet and is targeted at small atomic problems, such as identifying the color of an object. This ensures the **explainability** of the created tools. We can easily incorporate human efforts to examine the problematic tools and fix the errors. In addition, this allows for the decomposition of complex problems into multiple manageable steps, facilitating the **compositionality** of these created tools during inference.

# 3 EXPERIMENT

## 3.1 EXPERIMENTAL SETTING

**Evaluation Tasks, Datasets, and Metrics.** To demonstrate the versatility of CRAFT, we select three distinct tasks for evaluation, spanning visual question answering (VQA), tabular processing, and mathematical reasoning:

- **VQA**: The goal is to answer questions based on the information available in an associated image. We use three complex visual reasoning datasets, including GQA (Hudson & Manning, 2019), OK-VQA (Marino et al., 2019), and A-OKVQA (Schwenk et al., 2022). The GQA problems are more complex and require compositional reasoning to answer, while OK-VQA and A-OKVQA mainly use external real-world knowledge of objects or actions in the image. For evaluation, we formalize the VQA task as an open-ended generation problem and use the soft accuracy (SAcc) metric (Antol et al., 2015). In addition, we observe that LM-generated functions often produce descriptive responses instead of concise phrases, which hurts the exact match between predictions and ground-truth answers. This can potentially cause an underestimation of the performance, so we also use the F1 score for evaluation, which is frequently employed in extractive question-answering tasks (Rajpurkar et al., 2016).
- **Tabular Processing:** It evaluates an LLM's ability to process structured data in tables. We use TabMWP (Lu et al., 2023), a dataset with each sample containing one table and one corresponding problem in natural language. To handle the task, LLMs should understand the natural language descriptions of the problems, extract relevant information from the accompanying tables, and finally perform calculations based on the extracted information. We use the accuracy based on the exact match to measure model performance.
- **Mathematical Reasoning:** LLMs are expected to solve mathematical problems written in natural language, leveraging both their understanding of textual inputs and complex reasoning capabilities. We use the algebra subset of MATH (Hendrycks et al., 2021), containing 881 challenging competition-level algebra problems. Evaluating CRAFT on all subsets goes beyond our budget constraint but we believe CRAFT is equally applicable to other math problems. The models' performance is evaluated using accuracy.

**Baselines.** We compare CRAFT with baseline methods of four categories:

- **Basic Reasoning without Tools:** This line of methods solves downstream problems solely based on the intrinsic reasoning ability of LLMs without access to any external tool. We use the chain-of-thought prompting (**CoT**) (Wei et al., 2022), which prompts LLMs to generate the rationales before answers *without* using tools. However, it does not apply to the VQA task since LLMs cannot process visual information without external visual tools.
- **Tool Learning:** We compare with approaches that directly leverage existing tools to assist the problem-solving process. In this case, LLMs only learn to use the human-provided tools without creating and retrieving tools. We compare to two approaches: (1) **Vanilla** stands for utilizing the most basic tools, such as Python Interpreter for all three tasks, and extra vision models to solve VQA problems. Specifically, the vanilla tool-using method for VQA is ViperGPT (Sur'is et al., 2023), and that for the other two tasks is Program-of-Thoughts reasoning (Chen et al.,

Table 1: Distinctions between baseline methods and CRAFT in enhancing LLMs with created tools.

| Tool-Creation Method | Dataset for Create Tools | Reuse Tools? | Tool Base Size | Retrieval-enhanced? |
|---|---|---|---|---|
| CREATOR | Test Set | No | 0 | No |
| LATM | Train Set | Yes | 1 | No |
| CRAFT | Instruction Dataset or Train Set | Yes | >100; Theoretically Unlimited | Yes |

Table 2: The experimental results of CRAFT and four categories of baselines on three tasks. SAcc denotes soft accuracy, which is widely used for VQA. F1 is supplemented to tackle the issue of underestimated performance caused by the descriptive responses of LLMs. Acc denotes the accuracy.

| GPT-3.5-Turbo | | GQA | | OK-VQA | | A-OKVQA | | TabMWP | MATH$_{alg}$ |
|---|---|---|---|---|---|---|---|---|---|
| | Method | SAcc | F1 | SAcc | F1 | SAcc | F1 | Acc | Acc |
| Basic Reasoning | CoT | - | - | - | - | - | - | 75.2 | 50.9 |
| Tool Learning | Vanilla | 35.0 | 36.9 | 15.4 | 24.7 | 15.6 | 23.0 | 80.6 | 58.2 |
| | External | 34.2 | 37.8 | 16.8 | 25.3 | 14.5 | 22.9 | 83.1 | 41.1 |
| Different Tools | LATM | 29.4 | 30.3 | 7.8 | 11.8 | 6.5 | 11.4 | 9.3 | 30.3 |
| | CREATOR | 34.3 | 38.4 | 16.7 | 27.3 | 17.3 | 25.8 | 81.0 | 65.0 |
| Alternative Retrieval | SimCSE | 36.4 | 38.8 | 18.4 | 28.9 | 16.8 | 24.3 | 83.8 | 36.7 |
| | BM25 | 37.9 | 39.0 | 13.4 | 24.3 | 17.8 | 26.1 | **89.2** | 35.9 |
| This Work | CRAFT | **45.4** | **48.8** | **33.4** | **43.0** | **30.8** | **40.6** | 88.4 | **68.1** |

2022b). (2) **External library:** Therefore, we also explore the possibility of exploiting external tool functions in the Python libraries to enhance the vanilla methods. For VQA, we use Numpy (Harris et al., 2020), SciPy (Virtanen et al., 2020), Scikit-Image (Van der Walt et al., 2014), and Mahotas (Coelho, 2012). For the remaining two tasks, we substitute Scikit-Image and Mahotas with Pandas (McKinney et al., 2011) and SymPy (Meurer et al., 2017).

- **Different LLM-Created Tools:** We compare with previous tool creation approaches, including **LATM** (Cai et al., 2023) and **CREATOR** (Qian et al., 2023). Specifically, LATM samples 3 examples from the training set and applies GPT-4 to create a tool for the task, which is further verified by 3 samples from the validation set. The created tool is then applied to all test cases. CREATOR creates one specific tool for each test case in the inference time. For fair comparisons, we remove the format checking and rectifying process used in the original work and only measure the one-pass accuracy. The distinctions between these two methods and CRAFT are shown in Table 3.1.

- **Alternative Retrieval Methods:** We compare with previous tool retrieval approaches, which focus on the similarity measure between the problem and the API names. We include two prevalent measures, namely SimCSE and BM25 similarity, following Qin et al. (2023b) and Patil et al. (2023) respectively. The baseline retrieval methods are also based on our created toolset for fair comparison.

In this work, we implement CRAFT and all baselines based on the GPT-3.5-Turbo (ChatGPT) backbone because: (1) It is more cost-effective compared to alternatives like GPT-4, with affordable cost and strong performance; (2) The Turbo-0613 version is specially optimized for the tool-learning purpose. Conversely, alternative backbone models (e.g., CodeLlama (Rozière et al., 2023)) demonstrate near-random performance in our setting, which can be attributed to their suboptimal tool-using capabilities. The concrete implementation details are described in Appendix B.

## 3.2 Experimental Results

We present the results in Table 2. Particularly, we find that directly leveraging tools from external Python libraries fails to improve the performance, and in certain cases, may have a detrimental impact (e.g., in mathematical reasoning). This suggests that the relevance of tools affects the performance of augmented LLMs, motivating us to construct a high-quality tool base that customizes LLMs to each task. We observe that LATM struggles with all datasets and brings negative effects; CREATOR yields a notable enhancement in mathematical reasoning task performance, while its impact on other datasets appears marginal. This result suggests the necessity of sufficient and diverse tools to tackle problems of various categories in downstream datasets. For tool retrieval baselines, the performances vary across datasets. But in general, LLMs do not get substantial enhancement except on TabMWP, posing the need for better retrieval algorithms.

Table 3: Results of further analysis, encompassing ablation study on abstraction and retrieval components, as well as the comparison between ViperGPT and CRAFT with different backbones.

| GPT-3.5-Turbo | GQA | | OK-VQA | | A-OKVQA | |
|---|---|---|---|---|---|---|
| | SAcc | F1 | SAcc | F1 | SAcc | F1 |
| ViperGPT | 35.0 | 36.9 | 15.4 | 24.7 | 15.6 | 23.0 |
| CRAFT | **45.4** | **48.8** | **33.4** | **43.0** | **30.8** | **40.6** |
| w/o Abstraction | 37.1 | 39.7 | 31.0 | 41.4 | 28.0 | 39.3 |
| w/o Problem | 42.4 | 45.8 | 32.7 | 42.3 | 29.8 | 38.7 |
| w/o Name | 36.4 | 38.3 | 26.8 | 35.7 | 21.7 | 30.6 |
| w/o Docstring | 37.3 | 39.1 | 29.8 | 38.8 | 25.0 | 34.0 |
| GPT-4 | GQA | | OK-VQA | | A-OKVQA | |
| | SAcc | F1 | SAcc | F1 | SAcc | F1 |
| ViperGPT | 51.4 | 53.7 | 36.7 | 47.2 | 32.8 | 42.4 |
| CRAFT | **55.6** | **58.8** | **39.0** | **49.1** | **35.3** | **44.8** |

Overall, CRAFT demonstrates superior performance on all datasets, especially on the challenging VQA tasks. Significantly, CRAFT demonstrates a notable enhancement over the vanilla baseline, namely ViperGPT, with absolute SAcc improvements of 10.4, 18.0, and 15.2 observed on the GQA, OK-VQA, and A-OKVQA datasets, respectively. In addition, based on the same created toolset, the retrieval approach incorporated in CRAFT demonstrates overall better performance compared to alternative ones, which exhibit a certain level of performance variance. One exception is the comparison with BM25 on TabMWP. This discrepancy can be attributed to the presence of relatively straightforward patterns within this dataset, which do not sufficiently showcase the advantages of our approach in tool retrieval.

## 4  FURTHER ANALYSIS.

In this section, we conduct an in-depth analysis for CRAFT on VQA datasets. This task is particularly pertinent for assessing the impact of external tool augmentation, given that LLMs lack the capability to directly process images. Thus, it serves as a key testbed for measuring the influence of external tools.

### 4.1  DOES ABSTRACTION FACILITATE TOOL USE?

**Setup.** Abstraction is a crucial step in constructing the toolset, converting solutions for specific problems into general-purpose tools that are applicable to diverse problems with a common pattern. In this section, we explore its efficacy with an ablation study. To scrutinize this, we establish a control group, where the toolset is created ablating the abstraction step. To ensure compatibility, we prompt GPT-4 to assign a distinctive function name and docstring for each solution to facilitate the multi-view retrieval approach for fair comparison.

**Results.** Table 3 shows a clear performance drop when the abstraction step is ablated, confirming its importance. Moreover, comparing abstraction-ablated CRAFT with ViperGPT, improvements are achieved across all three datasets, especially on OK-VQA and A-OKVQA. We identify two potential reasons that can elucidate the improvement. First, the created toolset is large and diverse enough, facilitating the adoption of specific tools without abstraction for addressing new problems. Second, as retrieved tools offer a correct approach to problem-solving, LLMs can efficiently adapt these strategies to address new problems.

### 4.2  IS EVERY MATCHING IN THE RETRIEVAL TRIPLET EQUALLY IMPORTANT?

**Setup.** CRAFT retrieves tools based on multi-view matching. We demonstrate its effectiveness in Section 3.2. Next, we respectively ablate problems, function names, and docstring from the matching process to investigate their influence on performance.

**Results.** As demonstrated in Table 3, it is clear that the removal of any of the three similarity measures from our multi-view matching function adversely impacts performance, thereby validating

the rationale behind our design strategy. Among them, the function names appear the most important one, resulting in more than 6.6 absolute SAcc drop when ablated.

### 4.3 DOES CRAFT STILL WORK FOR MORE POWERFUL BACKBONE MODELS?

**Setup.** In previous experiments, CRAFT is implemented using GPT-3.5-Turbo as the backbone. In this analysis, we evaluate CRAFT when using the more powerful GPT-4 as the backbone. Due to the budget limits, we only compare CRAFT with the vanilla baseline ViperGPT without tool creation.

**Results.** The results in Table 3 demonstrate that CRAFT achieves consistently better performance with GPT-4, confirming that CRAFT is helpful even with more capable backbone models. However, it's noteworthy that while the improvement of CRAFT on GPT-4 is pronounced, it is less obvious compared to the impact on GPT-3.5-Turbo. We hypothesize that this result is in line with the conclusions of recent work, which finds that LLMs can benefit from the guidance of more capable models while gaining no improvement from the guidance of itself (Fu et al., 2023; Wang et al., 2023). The tools, created by GPT-4, may provide comparatively fewer insights for itself, thereby limiting the potential benefits of external tool augmentation.

### 4.4 CAN CRAFT IMPROVE PERFORMANCE AS THE TOOLSET GETS LARGER?

**Setup.** A feature of CRAFT distinctive from prior approaches is the extensibility of the toolsets. We examine the utility of extension by manipulating the toolset's size and tracking performance trends. To elaborate, the iterative problem sampling strategy detailed in Section 2.1 is initialized with a total of 11 epochs. In this analysis, the sizes of the toolset are modified through the exclusive inclusion of tools created at distinct epochs. We choose tools from the initial epoch, the final epoch, and the epoch in between, resulting in toolset sizes of 0 (no created tool for comparison), 261, 337, and 525, respectively.

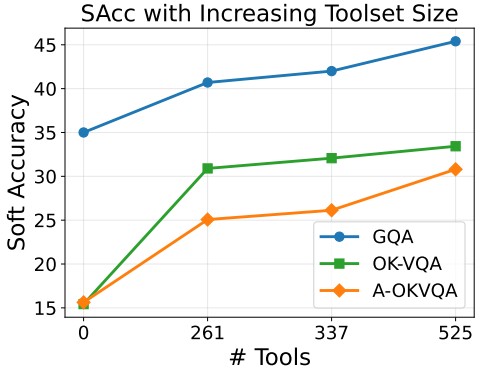

Figure 3: The performance of CRAFT improves as the toolset scales up.

**Results.** The results in Figure 3 show a consistent increase in soft accuracy as the toolset expands across 3 datasets, demonstrating the scalability and potential of CRAFT. The upward trend of soft accuracy continues, suggesting the potential for further improvement of CRAFT as the toolset keeps expanding. Significantly, the most substantial improvement is observed when transitioning from the absence of any created tools to the utilization of 261 tools. This validates the effectiveness of creating the specialized toolset to customize LLMs to various tasks and domains.

### 4.5 WHAT IS INSIDE THE TOOLSET?

We analyze the complexity and diversity of the code in toolsets. For complexity, we use the widely adopted cyclomatic complexity (Mc-Cabe, 1994) to measure the number of linearly independent paths, with the higher value indicating the code is more complicated and requires refactoring to make it more reliable. Good software should have a complexity of no more than

Table 4: Analysis of cyclomatic complexity and diversity of the toolsets.

| Task | VQA | Tabular Process | Mathematics Reasoning |
|---|---|---|---|
| Avg. Cyclomatic Complexity | 2.64 | 2.07 | 1.34 |
| # Tools | 525 | 181 | 282 |
| # Classes of Tools | 195 | 23 | 234 |

10, and a less complex toolset is desirable since it is less prone to trigger bugs. For diversity, we classify each tool into different groups. We use the number of distinct groups as the metric, with a larger number of tool groups indicating a wider range of problems that our toolset can address.

We calculate the complexity using Lizard Python library[2], and present the average complexity of tools for each task in Table 4. We observe that the created toolsets for 3 tasks exhibit relatively low complexity, indicating that the tools are well-structured and reliable. We then adopt the Louvain community detection method (Blondel et al., 2008), a graph-based community dividing algorithm, to group different tools. As shown in Table 4, for VQA, tabular process, and mathematics reasoning, there are 195, 23, and 234, distinct classes out of 525, 181, and 282 tools respectively. This suggests that the MATH dataset has the most diverse patterns, followed by VQA, while problems in the TabMWP dataset are more homogeneous and can be well-solved using fewer created tools.

## 5 RELATED WORK

### 5.1 TOOL LEARNING WITH LLMS

LLMs, when integrated with real-world Application Programming Interfaces (APIs), gain the capability to actively interact with a range of external systems (a.k.a, tools) (Parisi et al., 2022; Schick et al., 2023; Tang et al., 2023; Patil et al., 2023; Song et al., 2023; Hao et al., 2023; Wang et al., 2024). The pioneering work connects GPT-3 (Brown et al., 2020) with the web browser to access latest information, and hires human annotators to provide demonstrations of web searching (Nakano et al., 2021). Further research expands upon this concept by encompassing a broader range of tools, such as calculators, calendars, interpreter, physical simulator, and maps (Shuster et al., 2022; Paranjape et al., 2023; Liu et al., 2023c; Chen et al., 2022a; Gao et al., 2023a; Drori et al., 2022; Pan et al., 2023; Liu et al., 2023b), and explores the application of weakly-supervised methods, such as bootstrapping (Parisi et al., 2022; Schick et al., 2023). More recently, progress has been achieved through the process of distilling the tool using the ability of closed-source LLMs (ChatGPT (ChatGPT Plugins)) to the open-source LLMs. The key idea revolves around allowing ChatGPT to produce synthetic data exemplifying the usage of specified APIs. Subsequently, this synthetic data is leveraged for the refinement of open-sourced LLMs (Qin et al., 2023b; Tang et al., 2023). In this work, we extend our approach beyond mere dependence on existing tools. We adapt LLMs to diverse downstream tasks through the creation of customized tools and the retrieval of relevant tools during inference.

### 5.2 TOOL CREATION & RETRIEVAL

While the exploration on tool creation and retrieval is relatively limited compared to tool learning with LLMs, we identify some preliminary efforts in this domain. For tool creation, Cai et al. (2023) proposes an approach wherein tools are created through the utilization of three training samples, and their efficacy is subsequently assessed using three validation samples. Consequently, the resulting toolbase is constrained in quantity. This approach hinges on the assumption that there exists a notable similarity between the distributions of the training and testing data. Consequently, the tools produced can be readily incorporated. Similarly, Qian et al. (2023) adopts a strategy that involves generating tools exclusively based on the provided query. As a result, the created tools lack reusability, thereby undermining the fundamental purpose of tool creation. For tool retrieval, existing research primarily includes pre-selection of human-curated tools tailored to specific problems (Parisi et al., 2022; Tang et al., 2023; Schick et al., 2023; Zhuang et al., 2023), employing heuristic-based methods for tool selection (Shen et al., 2023; Liang et al., 2023), and adopting a straightforward similarity metric between user queries and API names (Qin et al., 2023b; Patil et al., 2023; Xu et al., 2023). In this work, we motivate to create a large tool base that can be effectively utlized on related downstream tasks and address the challenge of retrieving the relevant tools from the large tool base.

## 6 CONCLUSION

In conclusion, this paper presents CRAFT, a general framework for tool creation and retrieval to generalize LLMs for diverse domains and tasks. The framework's effectiveness is demonstrated through improved performance in challenging tasks, alongside insights into component contributions, constructed toolsets, and scalability.

---

[2]https://github.com/terryyin/lizard

ACKNOWLEDGEMENT

We thank the anonymous reviewers for their suggestions and comments. This research is based upon work supported by U.S. DARPA ECOLE Program No. HR00112390060 and U.S. DARPA ITM Program No. FA8650-23-C-7316. The views and conclusions contained herein are those of the authors and should not be interpreted as necessarily representing the official policies, either expressed or implied, of DARPA, or the U.S. Government. The U.S. Government is authorized to reproduce and distribute reprints for governmental purposes notwithstanding any copyright annotation therein.

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

APPENDIX

## A    LIMITATIONS AND FUTURE WORK

We identify two limitations in this work that are worth future exploration. First, although the basic idea in CRAFT is widely applicable in principle, it is currently based on code generation for tool creation. This indicates that CRAFT is only suitable for tasks that can be solved via writing code solutions. We plan to expand this scope by exploring the use of pseudocode to generalize CRAFT to more tasks. Second, the effectiveness of CRAFT is greatly affected by the tool-using ability of backbone models. In our pilot exploration, some open-source models achieve near-random performance in the challenging tool-manipulation setting. Future work includes eliciting the tool manipulation ability in open-source models, such as the pilot exploration in Qin et al. (2023b).

## B    CRAFT IMPLEMENTATION DETAILS

**Tool Creation.**   For VQA, we sample problems from general instruction datasets, constructing a comprehensive toolset without relying on strong supervised signals. We adopt LLaVA (Liu et al., 2023a), a substantial and diverse collection of vision-related queries, to facilitate the creation of a wide array of tools. Since LLaVA is designed in a conversational style, we construct another instruction dataset based on COCO-2017 (Lin et al., 2014) to complement problem diversity. The construction process follows the procedure in Chen et al. (2023), instantiating an interactive process to ask GPT-3.5-Turbo to generate a question-answering pair based on a caption of an image, followed by a filtering step. We prompt Turbo to make the answer concise instead of conversational. However, some questions generated in this manner are image-independent and can be answered by language models alone. Hence, we apply a filtering strategy by sending each question to Turbo and asking it if the question can be directly answered without images. We only select those samples that require visual data for answering. We sample 2,000 problems from the above instruction datasets, with 1,000 being from the primary random sampling epoch and another 1,000 from the subsequent 10 epochs, each contributing 100 problems per epoch. We employ ViperGPT (Sur'is et al., 2023) to generate specific code solutions. For tabular processing and mathematics reasoning, since there are no high-quality instruction datasets specified for these two tasks, we construct toolsets based on the training split of downstream datasets, i.e. TabMWP and MATH (algebra). For these tasks, we sample 500 problems from the above instruction datasets, 200 from the first random sampling epoch, and another 300 from the following 3 epochs, each yielding 100 problems per epoch.

**Tool Retrieval.**   In this work, we empirically set the number of retrieved tools $k$ to 10 for $q_t$, 5 for $f_t$, and 10 for $d_t$. Corresponding internal code implementations and docstrings will be incorporated into the prompt of the backbone model for subsequent utilization.

## C    PROMPT

### C.1    PROMPT TO FILTER IMAGE-DEPENDENT QUESTIONS IN VQA DATASET

```
You will be given a function named `llm_query`. The function Answers a
↪  text question using GPT-3 for reasoning and inference. Since GPT-3
↪  cannot process visual information, the question must be
↪  image-independent.
Then, you will be given a query. You need to decide if this llm_query
↪  function is able to **directly** solve this query. Directly answer
↪  yes or no.
Tips: If the query requires visual information of an image to solve, you
↪  should answer no. Otherwise, if the query is an image-independent
↪  inference task that can be solved by LLM reasoning or a search engine,
↪  you should answer yes.

Query: Why isn't Song Hee taking customers?
Answer: yes
```

```
Query: Why might one of the news anchors look angry, and the other look
↪  concerned?
Answer: no

Query: {query}
Answer:
```

## C.2    EVALUATE THE SPECIFIC GENERATION FOR VQA

```
Given the question for the visual question answering task: {question}
Does the following predicted answer have the same meaning as the
↪  reference answer in the context of the question?
Predicted Answer: {prediction}
Reference Answer: {reference}
You should compare the answers based on your understanding of the task,
↪  question, and answers, rather than relying on some superficial
↪  patterns like word overlap.
Directly answer Yes or No.
```

## C.3    TOOL CREATION

### C.3.1    VQA

```
**Rewriting Code to Create a Generic Tool Function**

**Purpose:** Given a query and its corresponding code solution, your task
↪  is to rewrite and abstract the code to create a general tool function
↪  that can be applied to similar queries. The tool function should
↪  solve a higher-level question that encompasses the original query and
↪  extend the code's functionality to make it more versatile and widely
↪  applicable.

Consider the following principles:

1. Understand the purpose of the query and infer a higher-level question
↪  that can be addressed by the tool function.
2. The generic tool function should solve queries of the same type, based
↪  on common reasoning steps rather than specific object types.
3. When enhancing the tool function's versatility according to the
↪  higher-level question, avoid assuming new attributes or methods of
↪  the `ImagePatch` classes.
4. Name the function honestly to ensure its functionality is not
↪  overclaimed.
5. Avoid using redundant and unused variables or intermediate steps in
↪  the tool function. Ensure that every step in the code has a purpose
↪  and directly contributes to the desired outcome.
6. Replace specific strings or variable names with general variables to
↪  enhance the tool's applicability to various queries.
7. Provide a docstring and an example of how to call the tool function to
↪  solve a specific query.
8. End your answer with the format 'The final generic tool with docstring
↪  is: ...' and 'The example to call the tool is: ...'.
---

**Example**
Query: Is there a backpack to the right of the man?
Specific code solution:
def execute_command(image)->str:
    image_patch = ImagePatch(image)
    man_patches = image_patch.find("man")
    if len(man_patches) == 0:
        # If no man is found, query the image directly with simple_query
        ↪  instead of returning a long string like "There is no man."
```

```
        return image_patch.simple_query("Is there a backpack to the right
        ↪  of the man?")
    man_patch = man_patches[0]
    backpack_patches = image_patch.find("backpack")
    if len(backpack_patches) == 0:
        return "no"
    for backpack_patch in backpack_patches:
        if backpack_patch.horizontal_center >
        ↪  man_patch.horizontal_center:
            return "yes"
    return "no"

Abstract tool:
The final generic tool with docstring is:
def check_existence_around_object_horizontally(image_patch: ImagePatch,
↪  object_name: str, reference_object_name: str,
↪  relative_horizontal_position: str, query: str) -> str:
    '''Check the existence of an object on either the left or right side
    ↪  of another object.

    Args:
        image_patch (ImagePatch): The image patch to check.
        object_name (str): The name of the object to check for existence.
        reference_object_name (str): The name of the reference object.
        relative_horizontal_position (str): The relative
        ↪  relative_horizontal_position position of the checked object
        ↪  to the reference object. Options: ["left", "right"].
        query (str): The original query to answer.

    Returns:
        str: "yes" if the object exists, "no" otherwise.
    '''

    assert relative_horizontal_position in ["left", "right"]
    reference_patches = image_patch.find(reference_object_name)
    if len(reference_patches) == 0:
        # If no reference object is found, query the image directly with
        ↪  simple_query instead of returning a long string like "There
        ↪  is no {reference_object_name}."
        return image_patch.simple_query(query)
    reference_patch = reference_patches[0]
    object_patches = image_patch.find(object_name)
    if len(object_patches) == 0:
        return "no"
    for object_patch in object_patches:
        if relative_horizontal_position == "left":
            flag = object_patch.horizontal_center <
            ↪  reference_patch.horizontal_center
        elif relative_horizontal_position == "right":
            flag = object_patch.horizontal_center >
            ↪  reference_patch.horizontal_center
        if flag:
            return "yes"
    return "no"

The example to call the tool is:
↪  check_existence_around_object_horizontally(image_patch, "backpack",
↪  "man", "right", "Is there a backpack to the right of the man?")

**Begin!**
Query: {query}
Specific code solution:
{solution}
```

Abstract tool:

## C.3.2 TABULAR PROCESSING

\*\*Rewriting Code to Create a Generic Tool Function\*\*

\*\*Purpose:\*\* Given a query and its corresponding code solution, your task
↪  is to rewrite and abstract the code to create a general tool function
↪  that can be applied to similar queries. The tool function should
↪  solve a higher-level question that encompasses the original query and
↪  extend the code's functionality to make it more versatile and widely
↪  applicable.

Consider the following principles:

1. The generic tool function should solve queries of the same type, based
↪  on common reasoning steps without mentioning specific object names or
↪  entity terms.
2. Name the function and write the docstring concerning both the core
↪  reasoning pattern and data organization format, without referencing
↪  specific objects.
3. Replace specific strings or variable names with general variables to
↪  enhance the tool's applicability to various queries. All columns
↪  names used inside the tool should be passed in as arguments.
4. Call the tool function to solve the original query. End your answer
↪  with the format '# The final generic tool with docstring is: ...' and
↪  '# The example to call the tool is: ...'.
---

\*\*Example\*\*
\*Table\*
Name: Orange candies per bag
Unit: bags
Content:
Stem | Leaf
2 | 2, 3, 9
3 |
4 |
5 | 0, 6, 7, 9
6 | 0
7 | 1, 3, 9
8 | 5
\*Query\*
A candy dispenser put various numbers of orange candies into bags. How
↪  many bags had at least 32 orange candies?
\*Specific code solution\*
```python
import pandas as pd
def count_bags_with_32_orange_candies(df):
    """
    This function takes in a pandas dataframe of orange candies per bag,
    ↪  and returns the number of bags that have at least 32 orange
    ↪  candies.
    Args:
    df (pandas.DataFrame): A pandas DataFrame object containing the
    ↪  number of orange candies per bag.
    The dataframe should contain "Stem" and "Leaf" columns.
    Returns:
    int: The number of bags that have at least 32 orange candies.
    """
    # prepare a list to calculate candies in each bag
    candies = []
    # calculate the total number of orange candies in each bag
```

```
        for i in range(len(df)):
            stem = df['Stem'][i]
            leaf = df['Leaf'][i]
            for j in range(len(leaf)):
                candies.append(stem * 10 + leaf[j])
        # filter the bags where the total number of orange candies is greater
        ↪   than or equal to 32
        filtered = [candy for candy in candies if candy >= 32]
        # count the number of items
        num_bags = len(filtered)
        return num_bags

data = {
    "Stem": [2, 3, 4, 5, 6, 7, 8],
    "Leaf": [[2, 3, 9], [], [], [0, 6, 7, 9], [0], [1, 3, 9], [5]]
}

df = pd.DataFrame(data)
count_bags_with_32_orange_candies(df=df)
```

Abstrcted tool function:
We're creating a generic tool function from specific code that counts the
↪   number of bags with at least a certain threshold of candies based on
↪   a stem-and-leaf plot. The original code combines the stem and leaf
↪   values to calculate the total number of candies in each bag, filters
↪   the bags with candies greater than or equal to the threshold value,
↪   and counts the number of such bags. We generalize the problem to
↪   create a flexible function for any stem-and-leaf plot of items and
↪   various threshold values. We replace specific columns, item names,
↪   and threshold values with generic variables like stem_col, leaf_col,
↪   item_threshold, and data_frame.
```python
# The final generic tool with docstring is:
def count_groups_above_threshold_in_stem_leaf(data_frame, stem_col,
↪   leaf_col, item_threshold):
    """
    This function takes in a pandas DataFrame representing a
    ↪   stem-and-leaf plot of groups and a threshold value, and returns
    ↪   the number of groups that have values greater than or equal to
    ↪   the threshold.

    Args:
    data_frame (pd.DataFrame): A pandas DataFrame containing the
    ↪   stem-and-leaf plot of items with columns specified by stem_col
    ↪   and leaf_col.
    stem_col (str): The column name for the stem values.
    leaf_col (str): The column name for the leaf values.
    item_threshold (int): The threshold value for filtering items.

    Returns:
    int: The number of items with values greater than or equal to the
    ↪   threshold.
    """
    # Initialize the list to calculate items in each group
    items = []

    # Calculate the total value of items in each group
    for i in range(len(data_frame)):
        stem = data_frame[stem_col][i]
        leaf = data_frame[leaf_col][i]
        for j in range(len(leaf)):
            items.append(stem * 10 + leaf[j])
```

```python
    # Filter the items where the total value is greater than or equal to
    ↪  the threshold
    filtered = [item for item in items if item >= item_threshold]

    # Count the number of items
    num_items = len(filtered)

    return num_items

# The example to call the tool is:
data = {
    "Stem": [2, 3, 4, 5, 6, 7, 8],
    "Leaf": [[2, 3, 9], [], [], [0, 6, 7, 9], [0], [1, 3, 9], [5]]
}

df = pd.DataFrame(data)
count_groups_above_threshold_in_stem_leaf(data_frame=df, stem_col="Stem",
↪  leaf_col="Leaf", item_threshold=32)
```

*Table*
pasta with meat sauce | $6.49
pasta with mushrooms | $9.05
spaghetti and meatballs | $7.43
mushroom pizza | $9.28
*Query*
How much money does Jayla need to buy 5 orders of pasta with meat sauce
↪  and 3 orders of pasta with mushrooms?
*Specific code solution*
```python
import pandas as pd

def calculate_total_cost(menu_df, orders):
    """
    This function takes in a pandas DataFrame representing a menu table
    ↪  and a dictionary of orders, and returns the total cost of the
    ↪  orders using pandas.
    Args:
    menu_df (pd.DataFrame): A pandas DataFrame containing menu items and
    ↪  their prices with columns 'Item' and 'Price'.
    orders (dict): A dictionary where the keys are menu item names and
    ↪  the values are the number of orders for each item.
    Returns:
    float: The total cost of the orders.
    """
    # Initialize the total cost
    total_cost = 0.0

    # Iterate through the menu items and calculate the cost for each
    ↪  ordered item
    for item, quantity in orders.items():
        # Filter the DataFrame for the specific item
        item_df = menu_df[menu_df['Item'] == item]
        if not item_df.empty:
            item_price = item_df['Price'].values[0]
            total_cost += quantity * item_price

    return total_cost

# Example usage:
menu_data = {
    'Item': ["pasta with meat sauce", "pasta with mushrooms", "spaghetti
    ↪  and meatballs", "mushroom pizza"],
    'Price': [6.49, 9.05, 7.43, 9.28]
}
```

```python
menu_df = pd.DataFrame(menu_data)

orders = {"pasta with meat sauce": 5, "pasta with mushrooms": 3}
calculate_total_cost(menu_df, orders)
```

Abstrcted tool function:
We're creating a generic tool function from specific code that calculates
↪  the total cost of items based on a table of item prices per unit and
↪  a dictionary of item quantities. We identify the core reasoning of
↪  the specific code is to calculate the total cost based on item prices
↪  and quantities for each item, i.e. total_cost = unit_price *
↪  item_quantity. The original code iterates through item names, filters
↪  the table for each item, and calculates the cost based on quantities.
↪  We generalize the problem to create a flexible function for any table
↪  of item prices and a dictionary of quantities. We replace specific
↪  columns and item names with generic variables like item_col,
↪  price_col, and item_quantities. We refactor the code with these
↪  variables, creating the new function
↪  calculate_total_cost_from_unit_prices_and_quantities.

```python
# The final generic tool with docstring is:
def calculate_total_cost_from_unit_prices_and_quantities(item_prices_df,
↪  item_col, unit_price_col, item_quantities):
    """
    This function takes in a pandas DataFrame representing a table of
    ↪  item prices and a dictionary of item quantities, and returns the
    ↪  total cost of the items based on the prices and quantities.

    Args:
    item_prices_df (pd.DataFrame): A pandas DataFrame containing item
    ↪  names and their prices.
    item_col (str): The column name for the item names.
    unit_price_col (str): The column name for the item prices.
    item_quantities (dict): A dictionary where the keys are item names
    ↪  and the values are the quantities of each item.

    Returns:
    float: The total cost of the items.
    """
    # Initialize the total cost
    total_cost = 0.0

    # Iterate through the item names and calculate the quantity for each
    ↪  item based on quantities
    for item_name, quantity in item_quantities.items():
        # Filter the DataFrame for the specific item name
        item_price_df = item_prices_df[item_prices_df[item_col] ==
        ↪  item_name]
        if not item_price_df.empty:
            item_price = item_price_df[unit_price_col].values[0]
            total_cost += quantity * item_price

    return total_cost

# The example to call the tool is:
item_prices_data = {
    'Item': ["pasta with meat sauce", "pasta with mushrooms", "spaghetti
    ↪  and meatballs", "mushroom pizza"],
    'Price': [6.49, 9.05, 7.43, 9.28]
}

item_prices_df = pd.DataFrame(item_prices_data)
```

```
item_quantities = {"pasta with meat sauce": 5, "pasta with mushrooms": 3}
calculate_total_cost_from_unit_prices_and_quantities(item_prices_df,
↪  "Item", "Price", item_quantities)
```

**Begin!**
*Table*
===table===
*Query*
===qst===
*Specific code solution*
```python
===specific solution===
```

Abstracted tool function:

### C.3.3 MATHEMATICS REASONING

**Rewriting Code to Create a Generic Tool Function**

**Purpose:** Given a table, query and its corresponding code solution,
↪  your task is to rewrite and abstract the code to create a general
↪  tool function that can be applied to similar queries. The tool
↪  function should solve a higher-level question that encompasses the
↪  original query and extend the code's functionality to make it more
↪  versatile and widely applicable.

Consider the following principles:

1. The generic tool function should solve queries of the same type, based
↪  on common reasoning steps without mentioning specific object names or
↪  entity terms.
2. Name the function and write the docstring concerning both the core
↪  reasoning pattern and data organization format, without referencing
↪  specific objects.
3. Replace specific strings or variable names with general variables to
↪  enhance the tool's applicability to various queries. All columns
↪  names used inside the tool should be passed in as arguments.
4. Call the tool function to solve the original query. End your answer
↪  with the format '# The final generic tool with docstring is: ...' and
↪  '# The example to call the tool is: ...'.
---

**Example**
Let \\[f(x) =\n\\begin{cases}\n3x^2 + 2&\\text{if } x\\le 3, \\\\\\nax − 1
↪  &\\text{if } x>3.\n\\end{cases}\n\\]Find $a$ if the graph of $y=f(x)$
↪  is continuous (which means the graph can be drawn without lifting
↪  your pencil from the paper).
Specific code solution:
```python
from sympy import symbols, Eq, solve

def find_a():
    """
    Finds the value of 'a' that makes the graph of the given piecewise
    ↪  function continuous.

    Returns:
    float: The value of 'a' that makes the graph continuous.
    """
    a, x = symbols('a x')
```

```
    # Define the piecewise function pieces
    left_side = 3*x**2 + 2
    right_side = a*x - 1

    # Find the value at the point of continuity (x = 3)
    left_value_at_3 = left_side.subs(x, 3)
    right_value_at_3 = right_side.subs(x, 3)

    # Create an equation to solve for 'a' based on the condition of
    ↪   continuity
    equation = Eq(left_value_at_3, right_value_at_3)

    # Solve the equation and return the value of 'a'
    solution = solve(equation, a)

    return solution[0]

find_a()
```
Abstrcted tool function:
assistant
We're creating a generic tool function from specific code that finds the
↪   value of a variable in a piecewise function that ensures continuity.
↪   The code calculates the appropriate value of a variable by ensuring
↪   that the left and right sides of the piecewise function at the point
↪   of continuity are equal. We generalize the problem to create a
↪   flexible function for any piecewise function, ensuring continuity at
↪   a specified point.
```python
from sympy import symbols, Eq, solve

# The final generic tool with docstring is:
def find_constant_for_continuity(x, constant, func1, func2,
↪   continuity_point):
    """
    Find the constant that makes a piecewise function continuous at a
    ↪   given point.

    Parameters:
    x (symbol): The symbol for the independent variable.
    constant (symbol): The symbol for the constant.
    func1 (function): The first piece of the piecewise function.
    func2 (function): The second piece of the piecewise function.
    continuity_point (float): The x-value where the two functions should
    ↪   be continuous.

    Returns:
    float: The constant value that makes the function continuous.
    """

    # Evaluate the functions at the point of continuity
    func1_at_cont_point = func1.subs(x, continuity_point)
    func2_at_cont_point = func2.subs(x, continuity_point)

    # Create an equation to solve for the constant based on the condition
    ↪   of continuity
    equation = Eq(func1_at_cont_point, func2_at_cont_point)

    # Solve the equation and return the value of the constant
    solution = solve(equation, constant)

    return solution[0]

# The example to call the tool is:
x, constant = symbols('x constant')
```

```
func1 = 3*x**2 + 2
func2 = constant*x - 1
continuity_point = 3
find_constant_for_continuity(x, constant, func1, func2, continuity_point)
```

Query: If $f(3)=1$ and $f(2x)=2f(x)$ for all $x$, find $f^{-1}(64)$.
Specific code solution:
```python
def find_inverse_value(base_value, base_result, target_result):
    """
    Finds the value of the inverse function based on the given properties
    ↪  of the original function.

    Args:
    base_value (float): The input value for the original function, f(x).
    base_result (float): The result of the original function,
    ↪  f(base_value).
    target_result (float): The value for which the inverse result needs
    ↪  to be found, f^{-1}(target_result).

    Returns:
    float: The value of the inverse function, x, such that
    ↪  f^{-1}(target_result) = x.
    """
    current_value = base_value
    current_result = base_result

    while current_result != target_result:
        current_value *= 2
        current_result *= 2

    return current_value

# Set the initial values and find the value of the inverse function
↪  f^(-1)(target_result)
inverse_value = find_inverse_value(base_value=3, base_result=1,
↪  target_result=64)
```
Abstrcted tool function:
We're creating a generic tool function from specific code that finds the
↪  value of an inverse function based on a set of rules for an original
↪  function. The original code solves for the inverse function by
↪  iterating over a multiplier rule and updating the corresponding
↪  values of the original function. We generalize the problem to create
↪  a flexible function for any linear combination rule with a common
↪  multiplier, a base input-output pair, and a target output value for
↪  the inverse function. We replace specific values and multipliers with
↪  generic variables like base_input, base_output, target_inverse_output,
↪  and common_multiplier.
```python
# The final generic tool with docstring is:
def find_inverse_value_linear(base_input, base_output,
↪  target_inverse_output, common_multiplier):
    """
    Find the value of the inverse function based on a linear combination
    ↪  rule for the original function.

    Parameters:
    base_input (float): The input value for the original function, f(x).
    base_output (float): The result of the original function,
    ↪  f(base_input).
```

```
        target_inverse_output (float): The output value for which the inverse
        ↪  result needs to be found, f^(-1)(target_inverse_output).
        common_multiplier (float): The common multiplier between the original
        ↪  function's input and output values.

        Returns:
        float: The value of the inverse function, x, such that
        ↪  f^(-1)(target_inverse_output) = x.
        """
        current_input = base_input
        current_output = base_output

        while current_output != target_inverse_output:
            current_input *= common_multiplier
            current_output *= common_multiplier

        return current_input

# The example to call the tool is:
find_inverse_value_linear(base_input=3, base_output=1,
↪  target_inverse_output=64, common_multiplier=2)
```

**Begin!**
Query: ===qst===
Specific code solution:
```python
===specific solution===
```

Abstrcted tool function:
```
```

## C.4  TOOL DEDUPLICATION

```
Here are several tools with similar functionalities. Your task is to
↪  select the most generic one, which can be widely applied and
↪  frequently reused across various scenarios. Your decision should be
↪  based on your understanding of typical use cases of VQA tasks and the
↪  capabilities of the tools, rather than relying on superficial
↪  patterns like the frequency of tool names.

Tips:
1. Consider the level of specificity of the strings in the code to assess
↪  its generalizability.
2. Evaluate the tool's functionalities and options to determine its
↪  versatility.

### Format ###
Tools are listed below:

No. 0:
Tool 1

No. 1:
Tool 2

...

No. N:
Tool N

Please respond with the numeric number preceding the most general tool
↪  using just one token, e.g.: N
```

```
### Input ###
The tools are:

{}

Please provide your answer by entering the numeric number preceding the
↪  most general tool using only one token:
```

## C.5 TOOL RETRIEVAL

### C.5.1 VQA

```
Given a query, convert it into a declarative command and then a brief and
↪  concise imperative instruction.
Next, infer tool functions that can be used based on the instruction.
Finally, infer the docstring of the tool functions.

Consider the following principles:
1. The instruction should reflect the action to take, rather than
↪  emphasizing specific noun phrases. So you should prioritize using
↪  general terms like `object`, `people`, and `action`, and so on,
↪  instead of directly saying specific names like `desk`, `american
↪  president`, and `stuffed animal`.
2. Use tool function names following the format `verb_noun` with less
↪  than five words. Consider utilizing the most frequently used words in
↪  function names listed below.
3. The docstring of the tool function should be general and abstract, not
↪  specific to the query. Consider utilizing the most frequently used
↪  words in function docstrings listed below.
4. End your answer with the format 'The useful functions are: [...]' and
↪  'The final answer is: ...', where '[...]' is a list of useful
↪  functions and '...' is the returned answer.
5. The most frequently used words in function names: ['object',
↪  'identify', 'check', 'objects', 'find', 'attribute', 'action',
↪  'location', 'determine', 'existence', 'infer', 'type', 'describe',
↪  'property', 'image', 'purpose', 'activity', 'count', 'interaction',
↪  'state']
6. The most frequently used words in function docstrings: ['specific',
↪  'object', 'identify', 'check', 'image', 'certain', 'given', 'another',
↪  'objects', 'find', 'type', 'existence', 'attribute', 'determine',
↪  'action', 'possible', 'list', 'two', 'infer', 'number']

Query: What is visible on the desk?
Let's think step by step:
First, the corresponding declarative command of the query is 'Identify
↪  the visible objects on the desk'. After abstracting, the general
↪  instruction should be 'Identify the objects on the specific
↪  surface.'.
So considering the naming rules of tool functions, the relevant and
↪  useful functions could be named as 'identify_objects' or
↪  'identify_objects_on_surface'.
Finally, we can infer that the docstring of the tool function could be
↪  'Identify the objects on the specified surface.'.
The useful functions are: ['identify_objects',
↪  'identify_objects_on_surface'].
The final answer is: Identify the objects on the specified surface.

Query: Which american president is most associated with the stuffed
↪  animal seen here?
Let's think step by step:
```

```
First, the corresponding declaritive command of the query is 'Search the
↪   american president most associated with the stuffed animal seen
↪   here'.\n\n"\
After abstracting, the general instruction should be 'Search people most
↪   associated with the specific object.'.\n\n"\
So considering the naming rules of tool functions, the relevant and
↪   useful functions could be named as
↪   'search_people_associated_with_object'.\n\n"\
Finally, we can infer that the docstring of the tool function could be
↪   'Search for people most associated with the specific object.'.\n\n"\
The useful functions are: ['search_people_associated_with_object'].\n\n"\
The final answer is: Search for people most associated with a specific
↪   object.

Query: {query}
Let's think step by step:
```

### C.5.2   Tabular Processing

```
Given a table and a corresponding query, please summary the task goal and
↪   briefly describe the row and column of the table.
Next, infer generic table processing tool functions that can achieve the
↪   task goal.
Finally, infer the docstring of the tool functions.

Consider following principles:
1. Generic tool function names should be less than eight words in length.
↪   Consider utilizing the most frequently used words in function names
↪   listed below.
2. The docstring should summaize the task goal and table format. Be
↪   general and abstract, not specific to the query. Consider utilizing
↪   the most frequently used words in function docstrings listed below.
3. End your answer with the format 'The useful functions are: [...]' and
↪   'The final answer is: ...', where '[...]' is a list of useful
↪   functions and '...' is the returned answer.
4. The most frequently used words in function names: ['count',
↪   'difference', 'items', 'total', 'value', 'calculate', 'frequency',
↪   'stem', 'leaf', 'groups', 'table', 'two', 'get', 'item', 'cost',
↪   'specific', 'entities', 'column', 'threshold', 'find', 'group',
↪   'unit', 'probability']
5. The most frequently used words in function docstrings: ['two',
↪   'number', 'item', 'column', 'items', 'specific', 'frequency',
↪   'values', 'name', 'total', 'difference', 'value', 'groups',
↪   'specified', 'table', 'given', 'row', 'stemandleaf', 'based', 'plot',
↪   'entities', 'target', 'names']

*Table*
Name: Orange candies per bag
Unit: bags
Content:
Stem | Leaf
2 | 2, 3, 9
3 |
4 |
5 | 0, 6, 7, 9
6 | 0
7 | 1, 3, 9
8 | 5
*Query*
A candy dispenser put various numbers of orange candies into bags. How
↪   many bags had at least 32 orange candies?
Let's think step by step:
```

```
To solve the problem, we should count the number of bags with at least a
↪  certain threshold of candies based on a stem-and-leaf plot. The code
↪  should combine stem and leaf values to calculate the total number of
↪  candies in each bag, filter the bags with candies greater than or
↪  equal to the threshold value, and count the number of such bags.
Considering the naming rules of tool functions, the relevant and useful
↪  functions could be named as
↪  'count_groups_above_threshold_in_stem_leaf' or
↪  'count_items_above_threshold_based_on_numeric_combination'.
Finally, we can infer that the docstring of the tool function could be
↪  'Given a threshold value and a pandas DataFrame representing a
↪  stem-and-leaf plot of groups, count the number of groups that have
↪  values greater than or equal to the threshold.'
The useful functions are: ['count_groups_above_threshold_in_stem_leaf',
↪  'count_items_above_threshold_based_on_numeric_combination'].
The final answer is: Given a threshold value and a pandas DataFrame
↪  representing a stem-and-leaf plot of groups, count the number of
↪  groups that have values greater than or equal to the threshold.

*Table*
pasta with meat sauce | $6.49
pasta with mushrooms | $9.05
spaghetti and meatballs | $7.43
mushroom pizza | $9.28
*Query*
How much money does Jayla need to buy 5 orders of pasta with meat sauce
↪  and 3 orders of pasta with mushrooms?
Let's think step by step:
To solve the problem, we should calculate the total cost of items based
↪  on a table of item prices per unit and a dictionary of item
↪  quantities. The code should calculate the total cost by using the
↪  formula total_cost = unit_price * item_quantity for each item,
↪  iterating through item names, filtering the table for each item, and
↪  calculating the cost based on quantities.
Considering the naming rules of tool functions, the relevant and useful
↪  functions could be named as
↪  'calculate_total_cost_from_unit_prices_and_quantities' or
↪  'calculate_total_quantity_from_items_and_coefficients'.
Finally, we can infer that the docstring of the tool function could be
↪  'Calculate the total cost of the items based on a pandas DataFrame
↪  representing a table of item prices and a dictionary of item
↪  quantities.'
The useful functions are:
↪  ['calculate_total_cost_from_unit_prices_and_quantities',
↪  'calculate_total_quantity_from_items_and_coefficients'].
The final answer is: Calculate the total cost of the items based on a
↪  pandas DataFrame representing a table of item prices and a dictionary
↪  of item quantities.

*Table*
{}
*Query*
{}
Let's think step by step:
```

### C.5.3 MATHEMATICS REASONING

```
Given a query, please infer the core mathematical skill for the solution.
Next, infer generic mathematical tool functions that can perform the core
↪  skill.
```

Finally, infer the docstring of the tool functions.

Consider the following principles:
1. Generic tool function names should be less than eight mathematic terms
  ↪   in length. Consider utilizing the most frequently used words in
  ↪   function names listed below.
2. The docstring should summarize the task goal. Be general and abstract,
  ↪   not specific to the query. Consider utilizing the most frequently
  ↪   used words in function docstrings listed below.
3. End your answer with the format 'The useful functions are: [...]' and
  ↪   'The final answer is: ...', where '[...]' is a list of useful
  ↪   functions and '...' is the returned answer.
4. The most frequently used words in function names: ['find', 'calculate',
  ↪   'sum', 'value', 'expression', 'difference', 'number', 'items',
  ↪   'total', 'time', 'target', 'inverse', 'generic', 'constant', 'max',
  ↪   'squares', 'proportional', 'product', 'consecutive', 'evaluate', 'x',
  ↪   'term', 'factor', 'largest']
5. The most frequently used words in function docstrings: ['number',
  ↪   'given', 'calculate', 'based', 'two', 'total', 'find', 'value', 'sum',
  ↪   'time', 'target', 'items', 'certain', 'numbers', 'amount', 'cost',
  ↪   'first', 'distance']

Query: Let $\\[f(x) =\n\\begin{cases}\n3x^2 + 2&\\text{if } x\\le 3,$
  ↪   $\\\\\nax - 1 &\\text{if } x>3.\n\\end{cases}\n\\]$Find $a$ if the
  ↪   graph of $y=f(x)$ is continuous (which means the graph can be drawn
  ↪   without lifting your pencil from the paper).
Let's think step by step:
To solve the problem, we should ensure the continuity of the function by
  ↪   equating the two function expressions at the boundary (x=3). The code
  ↪   should substitute x with 3 in both expressions, equate them, and
  ↪   solve for the unknown variable 'a'.
Considering the naming rules of tool functions, the relevant and useful
  ↪   functions could be named as 'solve_continuous_piecewise_function' or
  ↪   'find_constant_for_continuity'.
Finally, we can infer that the docstring of the tool function could be
  ↪   'Find the constant that makes a piecewise function continuous at a
  ↪   given point.'
The useful functions are: ['solve_continuous_piecewise_function',
  ↪   'find_constant_for_continuity'].
The final answer is: 'Find the constant that makes a piecewise function
  ↪   continuous at a given point.

Query: If $f(3)=1$ and $f(2x)=2f(x)$ for all $x$, find $f^{-1}(64)$
Let's think step by step:
To solve the problem, we should first understand the relationship between
  ↪   the given function and its inverse. Then, we need to use the provided
  ↪   information about the function and its properties to deduce the value
  ↪   of the inverse function at a given input. The code should analyze the
  ↪   properties to establish a connection (which is a linear relationship
  ↪   here) between the function and its inverse, and subsequently evaluate
  ↪   the result for the specific input.
Considering the naming rules of tool functions, the relevant and useful
  ↪   functions could be named as 'find_inverse_value_linear' or
  ↪   'find_inverse_value_based_on_properties'.
Finally, we can infer that the docstring of the tool function could be
  ↪   'Find the value of the inverse function based on a linear combination
  ↪   rule for the original function.'
The useful functions are: ['find_inverse_value_linear',
  ↪   'find_inverse_value_based_on_properties'].
The final answer is: 'Find the value of the inverse function based on a
  ↪   linear combination rule for the original function.

Query: {query}
Let's think step by step:

