# OpenReview forum: "CRAFT: Customizing LLMs by Creating and Retrieving from Specialized Toolsets"
_ICLR.cc/2024/Conference — ICLR 2024 poster_

### Official Review · Reviewer_3JvW · 2023-10-30

**Soundness:** 3 good
**Presentation:** 3 good
**Contribution:** 3 good
**Rating:** 8
**Confidence:** 3

**Summary:**

The paper presents CRAFT, a framework for tool creation and retrieval to customize large language models (LLMs) for various tasks and domains. CRAFT creates a specialized toolset by prompting LLMs to generate and abstract code solutions for problems, and validates and deduplicates the tools. CRAFT retrieves relevant tools from the toolset by multi-view matching, and adds them to the prompt of LLMs for code generation. CRAFT improves performance on vision-language, tabular processing, and mathematical reasoning tasks.

**Strengths:**

1. This paper proposes a tool generation and tool-using framework for LLMs which is a good attempt to enhance LLMs' capability to solve reasoning tasks by generating programs.

2. Personally I think the idea of a "pseudocode library" proposed in future work is cool and meaningful.

3. The experiments on baselines and different LLMs are comprehensive and the result is promising. Basically, I agree with the authors that tool generation puts a high demand on the LLMs' coding ability.

4. The created toolsets are a particularly important contribution to the LLM community.

**Weaknesses:**

1. The authors mentioned that alternative backbone models like CodeLlama demonstrate near-random performance. Can the authors provide such results (the performance of different LLMs in creating and using tools) in the experiment?

2. I suggest that the author should make the distinction between more specific methods more prominently in the main text (though the difference has been discussed in the experimental setting), such as by creating a table to compare various tool-augmented language model methods, and so on. The current Figure 1 appears to be similar to previous work like LATM, making it difficult to showcase the uniqueness of this article.

**Questions:**

1. What does "bug-free" mean and how do the authors ensure that the generated tools are "bug-free"?

2. What is the result of tool generation with GPT-3.5-turbo and other LLMs?

3. Can CRAFT be adapted to programming tasks like HumanEval since it generates "APIs"?

4. Can the authors discuss more on the "pseudocode library" like can we use a natural language library and how is it different from in-context learning?

5. Can the authors analyze more on the created toolsets like where they might it can be applied/generalized?

---

> ### Author Response · Authors · 2023-11-23
>
> We thank the reviewer for thinking our work makes an important contribution and also acknowledge the potential in our future work proposal. We are terribly sorry for giving late author responses because we are running some additional experiments, which consume a lot of time for autoregressive generation. The following is our response.
>
> Weakness
> 1. Can the authors provide such results (the performance of different LLMs in creating and using tools) in the experiment?
>
> | Model         | GQA   |       | OK-VQA |       | A-OKVQA |      |
> |---------------|-------|-------|--------|-------|---------|------|
> |               | Acc   | F1    | Acc    | F1    | Acc     | F1   |
> | CodeLLaMA-13B | 4.78  | 6.82  | 2.31   | 3     | 5.32    | 8.47 |
>
>
> In our experiments, we try CodeLLaMA-13B and observe that they can only achieve near-random performance on the visual tasks, as demonstrated in the table.
>
> 2. I suggest that the author should make the distinction between more specific methods more prominently in the main text.
>
> Thanks for your advice.  We summarize the distinctions between our work and previous work regarding tool creation in the following table:
> | Tool-Creation Method | Dataset for Create Tools | Reuse Tools? | Tool Base Size            | Retrieval-enhanced? |
> |----------------------|--------------------------|--------------|---------------------------|---------------------|
> | CREATOR              | Test Set                 | No           | 0                         | No                  |
> | LATM                 | Train Set                | Yes          | 1                         | No                  |
> | CRAFT                | Instruction Dataset or Train Set | Yes | >100; Theoretically Unlimited | Yes              |
>
> The major difference between CRAFT and LATM is that LATM uses three training samples to create only one tool for one dataset, and directly apply the tool to all downstream test samples, which is neither scalable nor generalizable. Hence, LATM cannot tackle tasks that contain diverse patterns. In contrast, with the help of our retrieval method, we are able to scale up the tool set and address various problems. Experiments in Table 1 have demonstrated that CRAFT achieves better performance.

---

> > ### Author Response · Authors · 2023-11-23
> >
> > Question
> >
> > 1. What does "bug-free" mean and how do the authors ensure that the generated tools are "bug-free"?
> >
> > Response: By “bug-free”, we mean that there is no (syntactic) bug in the curated tools and the execution results of the code match the ground truth reference. We ensure this through the “validation” process as stated in section 2.1, driven by a simple intuition of passing “test cases”: we offer GPT-4 access to the abstract tool function, with the expectation that it will address the original problems by supplying appropriate arguments to the tool function. The tools that fail to derive the correct answers given the original problems are discarded. We will clarify this in the revision.
> >
> >
> > 2. What is the result of tool generation with GPT-3.5-turbo and other LLMs?
> >
> > Response: We don't have quantitative results because we decided to use GPT-4 at an early stage right after thoroughly comparing the tool-generation quality of GPT-4 and GPT-3.5-Turbo, and we did not use Turbo to create any toolset thereafter. GPT-3.5-Turbo itself performs poorly on our benchmarks so both the generated specific solutions and abstracted tool functions have a much lower pass rate than GPT-4-generated tools in validation.
> >
> >
> > 3. Can CRAFT be adapted to programming tasks like HumanEval since it generates "APIs"?
> >
> > Response: Thanks for your comment. We’ve tried to include the code generation task. However, the problems in existing popular code datasets are usually atomic problems that may not be decomposed, and the ground-truth code snippets are already "tools" for the smallest unit (see humaneval: https://huggingface.co/datasets/openai_humaneval/viewer/openai_humaneval/test?row=0 ; mbpp: https://huggingface.co/datasets/mbpp/viewer/full/train?row=0). Therefore, retrieving tools for these problems resembles directly retrieving reference answers, which may not be a rigorous setup at inference time. Therefore, such characteristics of existing code datasets can be an obstacle to tool creation in code generation.
> >
> > 4. Can the authors discuss more on the "pseudocode library" like can we use a natural language library and how is it different from in-context learning?
> >
> > Response: The pipeline is the same as the current code-based CRAFT pipeline. For each training sample, we can generate a specific solution, which can be a chain-of-thought reasoning chain that derives the correct final answer. Then we can perform the abstract step to obtain a “lesson” or “experience” expressed in natural language from the specific solution. The “lesson” is then saved to the tool base, known as the “pseudocode library”. In the inference time, relevant “lessons” can be retrieved based on the given questions using the retrieval approach in the CRAFT framework.
> >
> > How is it different from in-context learning: In-context learning relies on existing training samples, comprising input-output pairs. Our tool creation approach enhances the generalizability of these training samples. This process involves distilling a “lesson” from the training sample, which encapsulates more broadly applicable and informative content than the original, unrefined training sample. The verification & deduplication steps in CRAFT can further improve the performance compared to in-context learning. In addition, CRAFT introduces a unique retrieving approach that can precisely pinpoint the related tools from the tool set, which is orthogonal to the in-context learning approach.
> >
> > 5. Can the authors analyze more on the created toolsets like where they might it can be applied/generalized?
> >
> > Response: We conduct some qualitative analysis of our created toolsets for the VQA problems. Due to the abstraction step in CRAFT, most created tools are atomic functions, aiming to tackle low-level problems that are generally existent. For example, the created tools include:
> >
> > identify_object_attribute: Identify the object located closest to a specific object
> >
> > identify_action: Identify the action of an object.
> >
> > find_object: Find the locations of a specific object.
> >
> > The created tools can work as a part of more complex visual problems. For example, in VQA problems shown in Figure 1, we extract several tools from the tool set but none of them can be applied to solve the problem directly. However, with the combination of two atomic and generic tools, “identify_objects_in_location” and “determine_object_location”, we can decompose the complex problem and then address it.

---

### Official Review · Reviewer_nWn7 · 2023-10-31

**Soundness:** 4 excellent
**Presentation:** 4 excellent
**Contribution:** 3 good
**Rating:** 6
**Confidence:** 3

**Summary:**

This paper introduces CRAFT, a novel framework designed to enhance large language models (LLMs) by creating and retrieving task-specific tools. CRAFT creates toolsets first and equips LLMs with a component that retrieves these tools to solve complex tasks. Experiments on vision-language, tabular processing, and mathematical reasoning tasks demonstrate the superiority of this approach over strong baselines. The analysis reveals that performance improvement is consistent when scaling the number of tools and the capability of backbone models, and that the created tools exhibit low complexity and atomicity.

**Strengths:**

1.	Traditional approaches to augment LLMs with tools lack flexibility, as they rely on general-purpose APIs. CRAFT addresses this problem by reusing task-related tools, which is more flexible.
2.	This method can adapt off-the-shelf LLMs to new domains and modalities without finetuning.
3.	Experiments show that the proposed framework can improve a lot compared to previous approaches.

**Weaknesses:**

1.	The setting of the experiments is a little bit limited. There are many agent benchmarks like MINT, AgentBench, and so on, which focus on the problem-solving capacity of LLMs as agents. The reviewer thinks the work needs to be further verified on broader benchmarks for agents.
2.	The comparison with LATM is a little bit unfair. The toolset created by CRAFT is the output of GPT-4, while the tool used by LATM is created by an inferior model if there is no misunderstood.
3.	The transferability of the toolset should be discussed as I noticed that the toolset for the VQA task and the toolset for the reasoning task are not the same. Maybe the authors can experiment to create a general tool set for all tasks and see what will happen.

**Questions:**

Please see the weakness.

---

> ### Author Response · Authors · 2023-11-23
>
> We thank reviewers for thinking that our approach is flexible to augment LLMs compared to traditional ones. The following is our response.
>
> Weaknesses
> 1. The setting of the experiments is a little bit limited.
>
> Response: Thanks for the advice. We did not include recent agent benchmarks, such as MINT and AgentBench, in our paper since they came out concurrently with CRAFT. We are also interested to see how CRAFT can benefit agents, but due to the time limits in the author response period, we are unable to adapt CRAFT to those benchmarks and finish the evaluation in time. However, since there is some overlap between our task and MINT/AgentBench, we expect the trends in our submission will translate to those in these two benchmarks.
>
> 2. The comparison with LATM is a little bit unfair.
>
> Response: In fact, for fair comparisons, CRAFT follows exactly the same setting as LATM, and creates tools using GPT-4 and uses tools with GPT-3.5-Turbo. We will highlight this in the revision.
>
> 3. The transferability of the toolset should be discussed.
>
> Response:
> Thanks for the suggestion. We have tried to mix the tools of different tasks into a single set and observe that this has little impact on the downstream task performance. We think this  can be attributed to the retrieval approach in CRAFT, which will only select tools that are most useful for the problem. We will include this in the revision.

---

### Official Review · Reviewer_A2gB · 2023-11-01

**Soundness:** 3 good
**Presentation:** 3 good
**Contribution:** 3 good
**Rating:** 6
**Confidence:** 3

**Summary:**

The paper introduces CRAFT, a novel framework for augmenting Large Language Models (LLMs) with specialized tools to tackle complex tasks. CRAFT generates task-specific toolsets and provides a retrieval component, enabling LLMs to offload functions to external modules through code snippets and APIs. This approach overcomes the limitations of general-purpose APIs, offering tailored solutions and improved flexibility. The framework ensures the quality and reusability of tools through validation, abstraction, and deduplication processes. Experiments across various domains, including vision-language, tabular processing, and mathematical reasoning, demonstrate substantial improvements over strong baselines. The paper's in-depth analysis confirms the scalability of CRAFT, the significance of each component, and the reliability and simplicity of the created tools. Ultimately, CRAFT presents a plug-and-play solution, enhancing the adaptability and problem-solving capabilities of off-the-shelf LLMs without requiring finetuning.

**Strengths:**

1. CRAFT showcases originality by combining tool learning, code generation, and retrieval to enhance LLMs' capabilities, applying this novel approach across various tasks and domains.

2. The framework is rigorously validated across different tasks, demonstrating substantial improvements and ensuring tool correctness and reusability, reflecting the high quality of the work.

3. The paper is well-structured and clearly written, providing a comprehensive presentation of the CRAFT framework, its applications, and experimental results.

4. CRAFT addresses crucial challenges in augmenting LLMs, offering a significant advancement in the field and demonstrating practical applicability and effectiveness across diverse domains.

**Weaknesses:**

1. The paper could benefit from a more detailed exploration of scenarios where CRAFT might not perform as expected. Understanding the limitations and potential failure cases of the framework would provide a more balanced view and help guide future improvements.

2. While the paper compares CRAFT to several strong baselines, expanding this comparison to include a wider range of existing tools and frameworks (e.g., SOTA methods in VQA) would strengthen the validity of the claimed improvements. This would also help in positioning CRAFT more clearly in the landscape of existing solutions.

3. The paper could provide a more in-depth analysis of the tool creation and retrieval components of CRAFT. Understanding how different types of tools contribute to performance improvements and how the retrieval mechanism interacts with various tasks would offer valuable insights.

4. While the paper mentions the scalability of CRAFT, providing empirical evidence and a more thorough discussion on how the framework scales with the number of tools and the complexity of tasks would be beneficial.

5. The paper could explore and address potential biases in the tool creation process, especially considering the reliance on GPT-4 for generating code solutions. Ensuring fairness and mitigating biases is crucial for the applicability of CRAFT across diverse scenarios.

6. Including a user study or examples of real-world applications of CRAFT could provide additional validation of the framework's practicality and effectiveness, offering a more comprehensive evaluation.

**Questions:**

1. Failure Cases: Can the authors provide specific examples or scenarios where CRAFT may not perform optimally? Insight into challenges or limitations faced by the framework would be valuable for a comprehensive understanding.

2. Baseline Comparison: Could the authors expand on the choice of baselines used for comparison? Including a broader range of existing tools and frameworks (existing SOTA methods) might help in better positioning CRAFT’s contributions.

3. Tool Creation and Retrieval Analysis: How do different types of tools contribute to the performance improvements observed with CRAFT? Additionally, how does the tool retrieval mechanism interact with various tasks?

4. Real-World Application: Are there examples of real-world applications where CRAFT has been applied? Including such examples or results from a user study could provide additional validation for the framework.

5. Tool Abstraction and Deduplication: Could the authors elaborate on the process of abstracting code solutions into reusable snippets and the criteria used for deduplication? Understanding this process in detail would provide clarity on the quality assurance of tools.

6. Handling of Descriptive Responses: The paper addresses potential issues with underestimated performance due to descriptive responses from LLMs. Could the authors provide more details on how this issue is handled or mitigated in CRAFT?

7. Scalability: The paper mentions the scalability of CRAFT. Could the authors provide empirical evidence or a more detailed discussion on how the framework scales with the number of tools and the complexity of tasks?

---

> ### Author Response · Authors · 2023-11-23
>
> We are glad the reviewer thinks CRAFT addresses crucial challenges in augmenting LLMs. The following are our authors' responses.
>
> Weaknesses
> 1. The paper could benefit from a more detailed exploration of scenarios where CRAFT might not perform as expected.
>
> Response: Thanks for the advice. We believe that such a systematic analysis is interesting, but during this response period, we failed to find an automatic method to identify and categorize them. Therefore, it may be beyond our capacity to finish the analysis in such a short time, but we will try to figure it out in the revision when time is available.
>
> We identify one specific failure mode in the CRAFT framework. In the abstraction process, there are some cases of successfully renaming the function names and writing a general docstring, and thus the tools look generalizable outside. However, inside the function, the specific variable names or textual inputs may fail to be converted into generic ones (expected results are shown in fig.2, replace all specific variable names with general ones, e.g., cat→animal, desk→object) and wrap textual inputs of internal function calls as arguments of the tool (e.g., date = df["date"]→date = df[column name], where the value of column name is passed in by tool users). Therefore, there are still some question-specific input-output formats. That said, the tool functions can only tackle a specific problem, instead of all problems of the same type. Such a mismatch between tool description and its implementation of the functionality could lead to errors when LLMs tend to call the tool to solve other problems. A potential solution from the tool side is to improve LLMs’ instruction-following ability so that they can abstract tools better, or introduce an extra verification process to filter out tools that “overclaim” their original functionality. In this case, we can further improve the quality of the created toolsets.
>
> 2.  Expanding the comparison to include a wider range of existing tools and frameworks.
>
> Response:
> | Method               | TabMWP | MATH |
> |----------------------|--------|------|
> | GPT-3.5-Turbo Vanilla| 68.2   | 25.7 |
> | POT+COT              | 80.0   | 54.0 |
> | POT+Rectify          | 81.2   | 63.8 |
> | POT+Rectify+COT      | 87.3   | 61.4 |
> | CRAFT                | 88.4   | 68.1 |
>
>
> We thank the reviewer for their constructive feedback. We have added four baselines on TabMWP and MATH. VQA, a third task we consider, takes more engineering efforts. As a result, we were not able to complete this experiment on VQA given limited time. Concretely, ‘GPT-3.5-Turbo Vanilla’ directly asks GPT-3.5-Turbo model to solve the problems; Program-Of-Thought (POT)+ Chain-Of-Thought (COT), on top of POT, adds COT prompting at the beginning; POT+Rectify introduces a rectification process when error occurs in POT programs; POT+Rectify+COT combines the above two techniques.
>
> Details on rectification: If an error occurs, then the LLM is prompted with demonstrations to correct the error. Applying a similar prompt format as before, the format of demonstrations “[EXAMPLE x]” now changes to “### Question [QST]\n ### Original [ORI]\n ### Error [ERR]\n ### Rectification [REC]”, where we provide the original tool implementation and calling decision in [ORI], offer the error tracebacks [ERR], and concatenate natural language reasoning on the error with the rectified code in [REC].
> In summary, both COT and the Rectify process can enhance POT on these two tasks, and their combination may further boost POT performance on TabMWP. However, CRAFT still outperforms all of them on both tasks.
>
> 3. The paper could provide a more in-depth analysis of the tool creation and retrieval components of CRAFT.
>
> Response: Thanks for the suggestion. For tool creation, how different types of tools contribute to performance improvements depends on what types of patterns exist in downstream tasks. As shown in our analysis section, there are tens of hundreds of types of tools in the toolset, so ablating different types of tools one by one would be extremely time-consuming and costly. Hence, we conduct a simpler experiment by removing the five most frequently retrieved tools from the MATH toolset and evaluating model performance on the reduced tool set. We found that the accuracy on MATH drops from 68.1 to 58.9, indicating that these tools are very useful to the task.
>
> For retrieval, the high-level strategy is task-agnostic: We prompt Turbo to “describe what it needs” by inferring the name and docstring of tool functions that might be helpful for each problem, and then retrieve tools that are most similar to the expected ones from our tool sets to solve the problem. However, to improve the precision of retrieval, we design task-specific prompts as shown in Appendix B.5, mainly characterized by providing a list of “most frequently used words” in the toolset so that Tuobo-inferred names can better match the tools in our sets.

---

> > ### Author Response · Authors · 2023-11-23
> >
> > Weakness
> > 4.  Providing empirical evidence and a more thorough discussion on how the framework scales with the number of tools and the complexity of tasks would be beneficial.
> >
> > Response: Thanks for your advice. In section 4.4, we have conducted experiments of enhancing GPT-3.5-Turbo with different numbers of tools and demonstrated that CRAFT improves the performance as the toolset gets larger. The accuracy improves as more tools are included, suggesting the potential for further improving CRAFT with more tools.
> >
> > 5. The paper could explore and address potential biases in the tool creation process.
> >
> > Response: Thanks for the advice regarding the exploration and mitigation of potential biases in the tool creation process. We acknowledge the critical importance of ensuring fairness and minimizing biases to enhance the applicability of CRAFT. Our approach to mitigating the biases includes the strategy of data scrutiny and filtering. We meticulously select appropriate datasets that are well-acknowledged to reduce the potential bias in the source datasets. In addition, we manually scan the created tools using the CRAFT framework. If the biased samples are involved, we will filter out the created tools to ensure the overall fairness of the created toolsets.
> >
> > 6. Including a user study or examples of real-world applications of CRAFT could provide additional validation of the framework's practicality and effectiveness.
> >
> > Response: Thanks for the great suggestion! Validating CRAFT in real-world applications is exactly what we have been working on right now. More specifically, we are in the process of combining CRAFT with the existing language model as agent frameworks, like AutoGPT, to enable LLMs tackle real-world problems by, e.g., creating new tools instead of relying on a fixed toolset. This is still work in progress and we are not able to get any concrete findings during the short response period. Should we have any, we will include the new results in the revision.

---

> > > ### Author Response · Authors · 2023-11-23
> > >
> > > Questions
> > >
> > > 1-4
> > >
> > > Response: Please refer to our responses to weaknesses 1,2,3,6 respectively.
> > >
> > > 5. Tool Abstraction and Deduplication: Could the authors elaborate on the process of abstracting code solutions into reusable snippets and the criteria used for deduplication?
> > >
> > > Response: Sorry for the confusion. We explain the detailed process in section 2.1.
> > >
> > > For abstraction, we instruct GPT-4 to replace all specific variable names with general ones (e.g., cat→animal, desk→object) and wrap textual inputs of internal function calls as arguments of the tool (e.g., date = df["date"]→date = df[column name], where the value of column name is passed in by tool users) within the code piece, substituting them with more generic counterparts to adapt to similar problems. Prompt can be found in Appendix B.3.
> > >
> > > For deduplication, we organize created tools into groups based on function names and the corresponding number of input arguments. Each group contains tools that have the same function names and the number of input arguments. For groups that contain more than one tool, we prompt GPT-3.5-Turbo to select the one that is most likely to be useful. Prompt can be found in Appendix B.4.
> > >
> > > 6. Handling of Descriptive Responses: The paper addresses potential issues with underestimated performance due to descriptive responses from LLMs. Could the authors provide more details on how this issue is handled or mitigated in CRAFT?
> > >
> > > Response: First, we find that LLM usually generates lengthy outputs with abundant words such as “the umbrella is pink”, while the ground truth in VQA datasets is usually concise, simply a word like “pink”. This discrepancy would hurt the exact match between predictions and ground truth answers. Hence, we adopt F1 scores to mitigate such issues from the metric side.
> > >
> > > Second, we find that Turbo tends to generate more descriptive responses than GPT-4. Hence, we apply GPT-4 to create tools to avoid such problems.
> > >
> > > 7. Scalability: The paper mentions the scalability of CRAFT. Could the authors provide empirical evidence or a more detailed discussion on how the framework scales with the number of tools and the complexity of tasks?
> > >
> > > Response: Please refer to our response to Weakness 4.

---

> > > > ### Comment · Reviewer_A2gB · 2023-11-23
> > > >
> > > > The authors respond to most of my questions/concerns. After reviewing the authors' responses and other reviews, I am keeping my initial score.

---

### Meta-Review · Area_Chair_DB56 · 2023-12-07

**Metareview:**

The paper presents CRAFT, a framework designed for creating and retrieving tools to customize LLMs across various tasks and domains. CRAFT enables the creation of a specialized toolset by prompting LLMs to generate and abstract code solutions for specific problems, followed by validation and deduplication of these tools. It retrieves relevant tools from this toolset through multi-view matching and incorporates them into the LLMs' prompts for code generation. CRAFT demonstrates enhanced performance in vision-language processing, tabular data handling, and mathematical reasoning tasks.

The paper is well written. The proposed method combines tool learning, code generation, and retrieval processes to augment the tool-use capabilities of LLMs. The task addressed is both critical and practical for many real-world applications.

Reviewers raised questions regarding its limitations, the need for comparisons with a broader range of existing tools and frameworks, empirical evidence supporting its scalability, comparisons with LATM, the transferability of the toolset, among other aspects. The authors have satisfactorily addressed these concerns in their rebuttal. Considering the overall positive ratings and the rebuttal's content, I recommend an accept (poster) rating.

**Justification For Why Not Higher Score:**

For a higher rating, further investigation into the transferability of the toolset would be beneficial. In their rebuttal, the authors mention their efforts to combine tools from different tasks into a single set, observing minimal impact on downstream task performance. However, in practical scenarios, task confusion frequently poses a significant challenge to achieving high performance. Therefore, clarifying this aspect is crucial.

**Justification For Why Not Lower Score:**

As mentioned above, the paper is generally well written. The proposed method combines tool learning, code generation, and retrieval processes to augment the tool-use capabilities of LLMs. The task addressed is both critical and practical for many real-world applications.

---

### Decision · Program_Chairs · 2024-01-16

Accept (poster)